# Public Attitudes toward Biobanking of Human Biological Material for Research Purposes: A Literature Review

**DOI:** 10.3390/ijerph16122209

**Published:** 2019-06-21

**Authors:** Jan Domaradzki, Jakub Pawlikowski

**Affiliations:** 1Laboratory of Health Sociology and Social Pathology, Karol Marcinkowski University of Medical Sciences, 60-806 Poznań, Poland; 2Independent Laboratory of Sociology of Medicine, Medical University of Lublin, 20-059 Lublin, Poland; jpawlikowski@wp.pl; 3Biobanking and Biomolecular Resources Research Infrastructure Poland, BBMRI.pl Consortium, 54-066 Wrocław, Poland

**Keywords:** biobanking, biobanks, donation, review, tissue banks, tissue donors, trust

## Abstract

*Background*: During the past few decades there has been a growing interest on the part of many governments in the creation of biobanks. Nevertheless, this would be impossible without participation of many donors who offer samples of their biological material for scientific research. Therefore, the aim of this paper is to provide an overview of the existing research on social attitudes towards biobanking. *Material and Methods*: A literature search was conducted in the database of MEDLINE (PubMed). 61 papers were included in the analysis. The retrieved articles were assessed using a thematic analysis. *Results*: Eight main themes were identified: (1) public knowledge about biobanks, (2) public views on biobanking, (3) willingness to donate, (4) donors’ motivations, (5) perceived benefits and risks of biobanking, (6) preferred type of consent, (7) trust toward biobanks, and (8) demographic characteristics of potential donors. *Conclusions*: Although the public lacks knowledge about biobanking, many individuals declare willingness to donate. Their will is influenced by: their knowledge about biobanking, the type of donated tissue, research purpose, concerns over the safety of the data, preferred type of consent, and trust towards biobanks.

## 1. Introduction

During the past few decades there has been a dynamic development of biobanks [1]. It was caused by the recognition of their potential in the field of public health, and by the hope that they will broaden knowledge about genetic, behavioral, and environmental determinants of many diseases, support the development of new drugs and diagnostic methods, and improve medical care toward more personalized medicine [2]. However, the functioning of any biobank requires constant participation of a large number of donors and building social trust toward research institutions. For this reason, it is crucial to know the attitudes of the public toward biobanks and factors influencing respondents’ willingness to donate.

As each biobank exists in a unique geographical, social, and historical context [3], donation is a complex process determined by people’s knowledge about biobanking [4,5], trust toward the government and research institutions [6,7,8], beliefs about the expected benefits [9], and donors’ cultural and religious beliefs on different types of tissues [10,11,12]. Consequently, knowledge about social attitudes toward biobanking may increase the effectiveness of the recruitment process [13,14]. This is important as the hype generated around biobanks may lead to an upsurge in exaggerated expectations and omission of possible risks [15].

Thus, the aim of this review is to identify, categorize, and analyse the main themes appearing in the existing research on social attitudes toward biobanking of human samples and data. It also discusses ways to improve social perception of and trust toward biobanks.

## 2. Material and Methods

A literature search was conducted in the database of MEDLINE (PubMed) using a combination of key words: ‘biobank’, ‘biobanking’, ‘tissue bank’, ‘donors’, ‘public opinion’, ‘social attitudes’, and ‘participation in the research’. To ensure the systematic aspect of the search, it was carried out twice: in November 2018 by J.D. and in January 2019 by J.P., and was limited to the material published after the year 2000.

The initial search identified 1161 publications, which were then selected on the basis of their titles and abstracts. Articles were included if they reported empirical studies on social attitudes toward biobanking, were written in English, and published in peer-reviewed journals. Papers were excluded if they focused on the theoretical aspects of biobanking or did not report on social attitudes toward biobanking. Comments, experts’ opinions, and letters to the editors were also excluded. These inclusion/exclusion criteria and additional reviews of the references of the selected articles yielded 61 articles which were read and analysed.

## 3. Results

Most research involved quantitative studies (*n* = 37) [7,9,10,16,17,18,19,20,21,22,23,24,25,26,27,28,29,30,31,32,33,34,35,36,37,38,39,40,41,42,43,44,45,46,47,48,49], while twelve were qualitative, including seven focus groups [6,12,50,51,52,53,54] and five structured interviews [11,55,56,57,58,59]. Six studies used mixed methods [4,8,11,60,61,62]. Additionally, six systematic research reviews were included [3,14,63,64,65,66].

Twenty-four studies were conducted in North America: 22 in the U.S. [4,6,12,16,17,19,20,23,26,28,29,30,34,43,45,46,48,51,53,54,57,62] and two in Canada [41,42]. Two studies described Pan-European studies [8,61], and 15 were conducted in different European countries: the UK [10,11], including Scotland [32,59], Sweden [18,21,26,27,60], Finland [35], the Netherlands [9], Italy [31,44,49], and Poland [37]. Ten studies were conducted in Asia: Jordan [38,39,40], Saudi Arabia [36,47], Singapore [22], China [7], Malaysia [67], India [58], and Japan [50]. Four studies were conducted in African countries: Nigeria [52], Uganda [24] and Egypt [33,55], and one in Australia [56]. As studies selected for the analyses were very heterogeneous, in terms of research design and goals (knowledge on biobanks, attitudes towards biobanking, preferred type of consent), methodologies (both qualitative and quantitative studies were included), the study populations (lay public, cancer patients, healthcare students, both developed and developing countries), different sample sizes, the type of biobank and biospecimens (DNA, genetic data, blood, residual specimens collected during the course of routine care), they provide limited insight into socio-empirical data and makes generalizing or comparing perspectives across different studies very difficult. Because this diversity does not allow for strict quantitative analysis, the results were integrated in an overall qualitative synthesis.

The retrieved articles were analysed using a thematic analysis [68]. The study results were selected by employing an inductive approach and classified as thematic categories, representing the views and attitudes of biobank participants. Due to a variety of methods used in the studies, both qualitative and quantitative findings were integrated in an overall qualitative synthesis. Eight main themes were identified: (1) public knowledge about biobanks, (2) public views on biobanking, (3) willingness to donate, (4) donors’ motivations, (5) perceived benefits and risks of biobanking, (6) preferred type of consent, (7) trust toward biobanks, and (8) demographic characteristics of potential donors. Nevertheless, not all articles comprised all these themes.

### 3.1. Public Knowledge about Biobanks

Although biobanks exist in many countries, a 2010 Eurobarometer study on biotechnology has demonstrated that two-thirds of Europeans have never heard about biobanks and less than 2% search for information about biobanking. A higher awareness was observed in Scandinavian countries, including Iceland (80%), Sweden (75%), and Norway (65%) [8]. At the same time, it should be stressed that the last European research was conducted almost a decade ago and social attitudes may have changed since that time. Nevertheless, later studies also showed that even the majority of Finns (83%) possess little knowledge about biobanks and 46% have never heard the term ‘biobank’ [35]. Infinitesimal knowledge on donation was also found in the UK [11]. While almost 84% of law, medical, and nursing students from Padua, Italy correctly understood the meaning of the term ‘biobank’, the majority did not know the difference between research and forensic biobanks, nor did they know about the existence of any biobank in Italy [49]. Although 72% of Polish students from the Faculty of Health Sciences of the Medical University of Białystok had heard about DNA biobanks, only 27% knew that DNA banking was conducted in Poland and none were able to name any city where biorepositories operate [37].

Similarly, up to 67% of Americans have not heard about biobanks [43] and many lack basic knowledge about biobanking [7,46]. The majority of Mexican-Americans have never heard the term ‘biobank’ [12]. Many respondents confuse participation in a biobank with medical examinations, i.e., diagnosis or treatment [64,65]. Only 25% of Jordanians had any knowledge about biobanks [38], and among Saudi students only 27% [47].

### 3.2. Public Views on Biobanking

Despite the deficits in knowledge, most research showed that public opinion on biobanking is generally positive and supports the idea of creating local biobanks. In Finland, 77% of respondents felt positive about such a project, while only 11% were against it [35]. In a Scottish survey, 82% of respondents positively evaluated the activity of biobanks [32], while 53.6% in Poland positively evaluated the activity of biobanks [37]. Between 84–98% of Americans believed that setting up a DNA databank was important or very important [4,28]. While only 25% of Jordanian population have heard about biobanking, 98% of respondents supported the idea of establishing a national biobank [39]. Similar results were found in Nigeria and Egypt [33,52]. Within the British population, 75% wanted to be asked for a donation and 87% thought it was important or extremely important [11]. Most respondents from Malaysia stressed the benefits over the risk emerging from biobanking and did not perceive donation as immoral [67]. At the same time, one must remember that comparing such different countries and heterogeneous populations is very difficult. Consequently, before making any generalizations and conclusions, involvement of more countries is needed.

### 3.3. Willingness to Donate

Better knowledge and positive opinions on biobanks correlate positively with respondents’ willingness to donate. In a Pan-European study, only 10% of respondents who had never heard about biobanks would not donate [61]. For instance, in Scandinavian countries, where the knowledge about biobanks is highest, 83% of Finns and 86% of Swedes declared such willingness [26,35], while only 4% of Greeks did [8]. On the other hand, out of 67% of American respondents who lacked knowledge about biobanks, 69% would donate [43], and among Saudi students surveyed 89% would donate [47]. In other studies, this percentage oscillated between 41% and 81%, although sometimes 25% of respondents would decline [29,33,45]. In the UK, almost 75% of respondents agreed with donation while 18% did not [10]. In China, it was 65% and 29%, respectively [7], in Saudi Arabia 81% and 47% [36], and in Jordan 64% and 33% [40]. It should be emphasized once more that, due to significant cultural differences between various research groups, such comparisons are of limited relevance.

Respondents’ willingness to donate was determined by the type of donated tissues, as the public were especially likely to donate blood, cancer, skin, and kidney tissues [58], but were less eager to donate eyes, brain, lungs, heart, bones, and germ cells left over from in vitro [10,11,26]. Respondents were also more prone to donate their blood (82%), saliva/sputum (77%), or urine (70%) than the organs of their deceased relatives (25%) [47]. Many declared donations for cancer research (85%), while research involving embryos (44%), combination of human samples with animals (34%), and research conducted abroad (35%) were much more controversial [11]. Cloning, stem cells research, and genetic engineering were also controversial [10,12,56]. Most respondents also objected to research with stigmatizing potential, i.e., on mental disorders, intelligence, homosexuality [16], or with presumed eugenic or commercial potential [4,6].

Respondents’ willingness to donate also depended on access to the information about the research, as many donors wanted to know who was conducting the research and where the research was being conducted, what was its purpose, who would have access to research results, and where and how the samples would be stored [6,12,51]. Donation may be further encouraged by the anonymization of samples [19,53] and by a possibility to withdraw from the research [42,47]. Other factors include positive recommendation by the bioethical committee [18,23,27], religious assent [40,52,67], and a conviction about the simplicity and safety of procurement of tissues [11,64].

In contrast, the donors may be discouraged by inadequate knowledge on biobanking [12,40,60], disapproval of the research [10,11,26], concerns over the safety of the data [30,48,59], fear over the invasive nature of the sampling procedure (pain, sight of blood, injections, and needles) [11,45], fear over infection with HIV [12], detection of genetic predispositions [22,44], and use of the sample in line with donors’ values [34,48,52]. Many respondents were afraid of stigmatization and discrimination [4,62], and commercial use of their samples [12,14]. Geographical distance from the biobank also discouraged some donors [11,44].

### 3.4. Donors’ Motivations

Most donors are driven by altruistic motives [26,45,58,59]. In most research, the most important motivation was a general feeling of duty [3,6,7,49,64] and desire to contribute to the common good [6,54,64]. Other important motives included helping others [34,55,57] and future generations [4,28,51]. Many wished to help generate new knowledge and develop new therapies [11,12,44,47]. Others expected benefits to their families, relatives or ethnic groups [43,54,64], or desired medical service and research results [53]. For example, in one American study, 81.1% of respondents declared eagerness to help others, 75.1% believed donation would increase knowledge on many diseases, and 61.1% hoped that research could help them or their families in the future [34]. In another study, 74% of respondents would donate to support scientific research [43]. In Sweden, the most important motivation was the need to help future patients (89%) and hope that research may benefit the donor’s or one’s family (61%). At the same time, 32.1% of respondents were motivated by a general sense of duty [26].

Although respondents propounded for the social benefits of biobanking over personal ones, they also believed that donation should benefit both parties [49]. Nevertheless, rarely did they expect financial gratification. Instead, they demanded information on the research results. For example, almost 89% of Americans wanted to be informed about the research results [19]. Among Hawaii natives the desire to be informed about research results was 88.9% [20,25], in the Netherlands 78% [9], in China 74% [7], in Egypt 89% [33], and in Uganda 48% [24]. In particular, the donors were interested in genetic research results [20,21]. For example, 66%-88% of the Dutch population wanted to be informed about genetic mutation [9]. Interestingly, 87% of Italians believed that the donors should not receive financial gratification [44].

### 3.5. Perceived Benefits and Risks of Biobanking

The most expected benefit resulting from biobanking was an increase in knowledge about many diseases and the development of novel therapies [6,12,14,55,58,59,62]. For example, 81.1% of American respondents believed that research conducted by biobanks could improve patient care and treatment of many diseases, while 75.1% stressed that it could help increase knowledge for society [34]. In another study, 68% of respondents stressed that donation is important because it contributes to research [43], allows doctors to research major diseases [28,53], and advances medical research and benefits society [47]. Also, Italian respondents stressed the utility of research conducted by biobanks and discovery of cures for major diseases [44,49]. At the same time, these benefits were primarily expected to affect future generations, respondents’ families, members of their ethnic group, and the donors themselves [45,48,54,64]. For instance, 89% of Swedish respondents stressed that biobank research would benefit future patients, and 61% hoped it would benefit themselves or their families [26]. Among Chinese respondents, 66.5% believed that donation would primarily benefit future patients [7], and many Australians hoped that donating their tissues might benefit their families [56]. The majority of outpatients from Maryland hoped that donations would improve their health (64%), their loved ones’ health (70%), or the health of someone of the same race or ethnicity (68%) [45,54,64].

Nevertheless, for the majority of respondents, participation in a biobank was a risky enterprise as they were afraid of the possibility of linking biological samples with donors’ personal data [8,26,29,30,34,52], and that the government, insurance companies, and employers could have access to such information, which might result in discrimination of the donors and their families [9,14,22,44,51]. Such risks were stressed by American outpatients who expressed concerns over possible discrimination (48%), privacy of data (36%), and were afraid of being used as guinea pigs (31%) [4,45]. Similar risks were also expressed by respondents in many European countries [8], Nigeria [52], and Singapore [22]. Additionally, respondents were also reluctant to the possibility of using their samples in research that was contrary to their values [10,11,12,14,25,46,53,56,57]. For example, many donors rejected a possibility of using their biological material for research involving female eggs (48%) or embryos (41%), or that done for commercial purposes [11]. For many others, research involving human cloning and genetic engineering, in general, were also unacceptable uses [10,12,56]. Finally, some respondents opposed research done for commercial profit [4,6,12,14]. Nevertheless, apart from these concerns, most respondents believed that the benefits outweigh the risks [22,51,54].

### 3.6. Preferred Type of Consent

Informed consent in the context of biobanking is a hotly discussed ethical and societal issue. Classic consent (specific or narrow, i.e., consent for one experiment with well-defined purposes, risks, and benefits) is not possible due to objective reasons—biospecimens are used in much research, by many scientists, working at different places. Therefore, new models of consent are proposed, such as blanket consent (it refers to a process by which individuals donate their samples without any restrictions), broad consent (refers to a process by which individuals donate their samples for a broad range of unspecified future studies with some restrictions), dynamic consent (a digital decision-support where modern IT communication strategies are used to continuously inform and offer choices to donors to specify the types of research for which their specimens can be used or not), or tiered consent (research can be subdivided into tiers or categories and participants can specify the types of research for which their specimens will be used). Although, from a biobank’s perspective, broad consent is preferable [69], it is not optimal for the donors. Thus, while some studies indicate respondents’ preferences for broad consent [12,29,38,43,45], others show that, if available, other options are preferable [4]. For example, in Europe, general consent was more accepted in Scandinavia (33–42%) than in southern countries: Bulgaria, Greece, and Turkey (11–16%), while 67% of the European population opted for narrow consent while and only 24% preferred broad consent [8]. Moreover, most Swedes rejected the possibility of conducting any type of research without explicit consent of the donors (62%), and 22.3% wanted to re-consent for every new research purpose [18]. Also, 42% of Finns expected re-consent, if new research differs from the original, and 29% before any new research [35]. Among Canadian leukemia patients, 60% preferred broad consent, but 30% chose tiered consent, and 10% preferred specific consent [42]. In another American study, 44% of respondents perceived broad consent as unacceptable and 38% as the worst option. Interestingly, for 43% of Americans, specific consent was also unacceptable, and for 45% it was the worst type of consent [46]. Thus, while the donors reject both extremes, they want to preserve some control over the donated samples [14,66].

### 3.7. Trust toward Biobanks

Trust toward biobanks correlates positively with the willingness to donate, preference for broad consent, and decreases perception of the risks related to the privacy and confidentiality of samples [12,48,58,61]. At the same time, most respondents trusted scientific institutions more than commercial, governmental, or insurance institutions [5,9,21,27,42,59,66]. For instance, 92% of American respondents trusted academic and medical researchers, while 80% trusted government researchers, and 75% pharmaceutical company researchers [30]. Also, Canadians trusted academic researchers (45%) more than those financed by the private sector (19%) or private biobanks (6%) [41]. Among Scottish respondents, 97% trusted university hospitals, while only 6% for-profit organizations [32]. Similarly, the majority of Chinese respondents trusted hospital research institutions (37.7%), the Chinese medical association (34.6%), and government institutions (30.3%), while only 2% trusted for-profit research companies [7]. Additionally, more people trusted national institutions (61–64%) than foreign biobanks (37–38%) [33,35].

Interestingly, lack of trust was significantly high among ethnic minorities: African-Americans, Mexican-Americans, Native Americans, Hawaii and Alaskan Natives, which resulted from their negative experiences with colonization, eugenics, and medical experiments [23,25,54,57].

### 3.8. Demographic Characteristics of Potential Donors

Some studies suggest that middle-aged (usually 40–65 years old) persons and older are more favorable toward donation, trust biobanks, and accept broad consent more often [11,26,29,30,43]. For example, a study by Lewis et al. showed that respondents over 55 years of age were more eager to donate [11], while in another research conducted by Goddard et al. almost 70% of persons over forty were in favor of donation. In contrast, only 31% of those under that age would donate [29]. On the other hand, some studies suggest that individuals aged under fifty were more in favor of specific consent [4,27] and that increasing age reduces the number of respondents willing to donate [7,32,38]. People with higher education are more eager to donate, but also expect re-consent [8,17,38].

Positive attitudes toward donation were more common among respondents with higher economic status [29,40], who lived in urban areas [7], and had children [26,36]. Those with a lower socio-economic status expected a possibility to withdraw samples and re-consent more often [4,40]. In some studies, males were more eager to donate than females [29,35] and accepted broad consent more often [8,30]. In contrast, Saudi and Egyptian females were more willing to donate [33,36].

Ethnic minorities were generally less eager to share their samples than people of European ancestry. They also expect re-consent more often [20,23,48].

While religious beliefs did not seem to influence donors’ decisions [22,28], in the British study, nonbelievers and less religious persons were more interested in donation [11,48]. Nevertheless, religious acquiescence may be an important motivator [40,52]. In Malaysia, Christians perceived more benefits and threats resulting from biobanking than Hindus did [67].

## 4. Discussion

This review of the research confirms that, although a large part of respondents do not possess knowledge about biobanking [37,61], many respondents are willing to donate their biospecimen, as their readiness to participate in the biobank depends, not only on respondents’ knowledge [5,8], but also on their declared system of values [12,34,52], experiences with healthcare system [64], trust toward the government and scientific institutions [7,26], their beliefs toward the benefits and risks associated with biobanking [9,30], and sociodemographic characteristics [10,22]. Nevertheless, it is a mistake to focus on any particular factor as they all are interrelated.

It should also be emphasized that different attitudes toward biobanking result from social, cultural, and religious variances, which determine what types of tissue people are ready to donate, the type of research they are eager to participate in, and the consent model they prefer. Thus, while planning a biobank, it is crucial to address these socio-cultural circumstances, as it warrants respect for the donors and ensures the success of the recruitment process [3].

Thus, active engagement of the donors in a biobank’s activity should not be viewed as an obstacle, but as a factor that enables their recruitment. Consequently, many authors argue that the organization of any biobank requires building a unique culture of trust, which should include: transparency of the biobank’s activity, appreciation of the donors, active involvement of local communities in planning and control of biobank activity, strengthening of bioethical committees in the organization and supervision of biobanks, and mutual communication with individual donors. Thus, recommendations include that donors have to have a chance to express their expectations and fears, receive clear and communicable leaflets, and feel in control (voluntary participation, possibility of withdrawal, new models of informed consent, e.g., dynamic consent). Further strategies include using the media of mass communication, including the Internet and social media; communication with representatives of patients’ organizations, local community and other stakeholders; promotion of active participation and engagement of the donors in promoting the idea of biobanking; access to up-to-date information on a biobank’s research and its results; contact with researchers; access to research results; and references to common good and altruism, taking care of the cultural and religious diversity of the donors [3,27,56]. These recommendations are of special significance in countries, like Poland, which has launched a project aiming at the organization of a Polish network of biobanks [70].

Although this study brings new insight into the public attitudes toward biobanking of human biological material for research purposes, it also has some limitations. Searching was limited to one database and some studies could not be identified, but this limitation should not change the general view and conclusions. Moreover, as analysed studies were conducted with different populations, it is hard to make quantitative comparisons. However, qualitative analysis is still possible and justified. In the future one should strive for strict quantitative analysis of public attitudes towards biobanking.

## 5. Conclusions

While some limitations may exist in this paper, some advantages should also be acknowledged. Our review indicates that, although the majority of respondents lack basic knowledge about biobanking, many are open to donation and support the idea of establishing biobanks. Willingness for donation is influenced by multiple socio-cultural factors, including: the knowledge about biobanking, the type of donated tissue, research purpose and ethical standards, concerns over the safety of the data, positive recommendation by the bioethical committee, and commercial or non-scientific use of their samples. What is equally important is that most of the donors are driven by altruistic motives. Another notable finding is that respondents fear linking biological samples with their personal data, and access to their sensitive data by the government, insurance companies, and employers, and, consequently, discrimination or stigmatization. In particular, they are afraid of using their samples in research contrary to their values. Thus, although many donors accept broad consent, if available, other options are preferable, i.e., dynamic consent. This review also shows that the public trust public and national biobanking institutions rather than commercial and foreign institutions; trust toward biobanks links positively with willingness to donate, preference for broad consent, and links negatively with concerns about privacy protection and being a member of ethnic minorities. Biobankers who establish and manage their biobanks should take into account socio-cultural circumstances and care about a culture of trust towards biobanks, research, and scientists.

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
