# Peer review of "Public Attitudes toward Biobanking of Human Biological Material for Research Purposes: A Literature Review"

_ijerph, 2019, doi:10.3390/ijerph16122209_

Round 1
Reviewer 1 Report
comments to the manuscript:
Material and Methods:
49 … search was carried out twice in November 2018 and January 2019
50 and was limited to the material published after the year 2000.
Why was the search done twice, there is no explanation
51…search identified 1161 publications which were pre-selected
56 … 61
57 articles which were read and analyzed
This means only 5.25% of ALL articles were taken in the consideration! It is a small number for such a great study. The authors explain it by a great heterogeneity and would be interesting to consider the journals where these articles were published, please complete it
88 Although biobanks exist in many countries, a 2010 Eurobarometer study…
I recommend more actual data no 9 years old
105 3.2. Public Views on Biobanking
It is difficult to compare the situation in a few and so different countries like USA, Nigeria, Jordan, I suggest commenting on it that it needs more countries to be involved
173 3.5. Perceived Benefits and Risks of Biobanking
Many cited/refered literature sources but without an appropriate explanation, please try to be more precise
185 3.6. Preferred Type of Consent
Many types of informed consent (IC) mentioned: Blanket, broad, dynamic, tiered, general, narrow, all these IC must be explained and cited properly
223 3.8. Demographic Characteristics of Potential Donors
Middle-aged, older persons young, please specify especially for „young“ there are different „age“ cathegories, in some countries it is 16, in others 18,…
References:
In following references, I recommend revisions:
5. why „ithalics“, the same 20., 63.,
24. must be revised
48., 66. how many authors do you have to mention?
Author Response
Response to Reviewer 1 Comments
Point 1: Material and Methods:
49 … search was carried out twice in November 2018 and January 2019
50 and was limited to the material published after the year 2000.
Why was the search done twice, there is no explanation
Response 1: We explain that the aim of double search was to ensure the systematic aspect of the search and to complement the literature:
“To ensure the systematic aspect of the search, it was carried out twice: in November 2018 by J.D. and in January 2019 by J.P. and was limited to the material published after the year 2000.”
Point 2:
51…search identified 1161 publications which were pre-selected
56 … 61
57 articles which were read and analyzed
This means only 5.25% of ALL articles were taken in the consideration! It is a small number for such a great study. The authors explain it by a great heterogeneity and would be interesting to consider the journals where these articles were published, please complete it
Response 2: We have written incorrectly that the identified 1161 publications were pre-selected while in fact these papers were afterwards selected on the basis of theirs titles, abstracts and including/excluding criteria. The study sample covers only 5.25% of all articles identified, but in fact the vast majority of publication identified did not focus on social attitudes toward biobanking or addressed theoretical, including legal or ethical aspects of biobanking. We have corrected this paragraph as follows:
“The initial search identified 1161 publications which were then selected on the basis of their titles and abstracts. Articles were included if they reported empirical studies on social attitudes toward biobanking, were written in English and published in peer-reviewed journals. Papers were excluded if they focused on the theoretical aspects of biobanking or did not report on social attitudes toward biobanking. Comments, experts’ opinions and letters to the editors were also excluded”.
Point 3:
88 Although biobanks exist in many countries, a 2010 Eurobarometer study…
I recommend more actual data no 9 years old
Response 3: While we acknowledge the Reviewers’ comment, we would like to note that we began this paragraph by citing the 2010 Eurobarometer study on biotechnology because it is the latest European report on biotechnologies. While there are some other European studies on biotechnologies, all of them were conducted before 2010 (Biotechnology SP341 [2010], Europeans and biotechnology: patterns and trends SP244b [2005], Europeans and biotechnology SP177 [2002], Europeans and biotechnology SP134 [1999], just to mention a few, see: An alphabetical guide to Eurobarometer surveys: file:///C:/Users/user/Desktop/EB_ListA-Z_070619.pdf) and not all of them addressed biobanks. Moreover, while the European Commission had published yet another report in 2011: Biobanks for A Challenge for Governance Europe, it mainly summarized the results from the studies by Gaskell et al. we cite in our paper (8 and 61). On the other hand, the Biotech Barometer (H1 2018) that was carried out in 2018 focused solely on trends and figures of European biotech companies and did not refer to biobanks.
Thus, we have started by citing this European study to give the reader a broader perspective and then we give some examples from different studies conducted in various European and non-European countries. Moreover, later in this paragraph we cite both older studies [Tupasela et al 2010; Krajewska-Kułak et al. 2011; Ahram et al. 2012; Ma et al. 2012; Lewis et al. 2013; Gaskell et al. 2013; Rahm et al. 2013 and Nobile et al. 2013] and more updated research [D’Abramo et al. 2015; De Vries et al. 2016; Tozzo et al. 2017; Merdad et al. 2017 and Heredia et al. 2017].
Nevertheless, acknowledging the Reviewers’ suggestion we have added a comment:
Although biobanks exist in many countries, a 2010 Eurobarometer study on biotechnology has demonstrated that two-thirds of Europeans have never heard about biobanks and less than 2% search for information about biobanking. A higher awareness was observed in the Scandinavian countries, including Iceland (80%), Sweden (75%) and Norway (65%) [8]. At the same time, it should be stressed that the last European research was conducted almost a decade ago and social attitudes may have changed since that time. Nevertheless, also later studies show that even the majority of Finns (83%) possess little knowledge about biobanks and 46% have never heard the term ‘biobank’ [35].
Point 4:
105 3.2. Public Views on Biobanking
It is difficult to compare the situation in a few and so different countries like USA, Nigeria, Jordan, I suggest commenting on it that it needs more countries to be involved
Response 4: While in paragraph 3.3. Willingness to Donate (lines 126-127) we stress that due to significant cultural differences between various research groups any comparisons are of limited relevance we agree that such comment should have appeared also earlier. Thus, we have added a comment suggested by the Reviewer.
Despite the deficits in knowledge, most research show that public opinion on biobanking is generally positive and supports the idea of creating local biobanks. In Finland, 77% of respondents felt positive about such a project, while only 11% were against it [35]. In a Scottish survey, 82% of respondents positively evaluated the activity of biobanks [32] and in Poland 53.6% [37]. 84-98% of Americans believed that setting up a DNA databank was important or very important [4,28]. While only 25% of Jordanian population have heard about biobanking, 98% of respondents supported the idea of establishing a national biobank [39]. Similar results were found in Nigeria and Egypt [33,52]. 75% of British population wanted to be asked for a donation and 87% thought it was important or extremely important [11]. Most respondents from Malesia stressed the benefits over the risk emerging from biobanking and did not perceive donation as immoral [67]. At the same time, one must remember that comparing such different countries and heterogeneous populations is very difficult. Consequently, before making any generalizations and conclusions involvement of more countries is needed.
Point 5:
173 3.5. Perceived Benefits and Risks of Biobanking
Many cited/refered literature sources but without an appropriate explanation, please try to be more precise
Response 5: Entire paragraph has been revised:
The most expected benefit resulting from biobanking was an increase in knowledge about many diseases and the development of novel therapies [6,12,14,55,58,59,62]. For example, 81.1% of American respondents believed that research conducted by biobanks could improve patient care and treatment of many diseases, while 75.1% stressed it could help increase knowledge for society [34]. In another study, 68% of respondents stressed that donation is important because it contributes to research [43], allows doctors to research major diseases [28,53], advances medical research and benefits society [47]. Also, Italian respondents stressed the utility of research conducted by biobanks and discovery of cures for major diseases [44,49]. At the same time, these benefits were primarily expected to affect future generations, respondents’ families, members of their ethnic group and the donors themselves [45, 48, 54, 64]. For instance, 89% of Swedish respondents stressed that biobank research would benefit future patients and 61% hoped it would benefit themselves or their families [26]. 66.5% of Chinese respondents believed that donation would primarily benefit future patients [7] and many Australians hoped that donating their tissues might benefit their families [56]. The majority of outpatients from Maryland hoped that donations would improve their health (64%), their loved ones (70%) or someone of the same race or ethnicity (68%) [45,54,64].
Nevertheless, for the majority of respondents participation in a biobank was a risky enterprise as they were afraid of the possibility of linking biological samples with donors’ personal data [8,26,29,30,34,52] and that the government, insurance companies and employers could have access to such information, which might result in discrimination of the donors and their families [9,14,22,44,51]. Such risks were stressed by American outpatients who expressed concerns over possible discrimination (48%), privacy of data (36%) and were afraid of both being used as guinea pigs (31%) [45, 4]. Similar risks were also expressed by respondents in many European countries [8], Nigeria [52] and Singapore [22]. Additionally, respondents were also reluctant to the possibility of using their samples in research that was contrary to their values [10-12,14,25,46,53,56,57]. For example, many donors rejected a possibility of using their biological material for the research involving female eggs (48%) or embryos (41%) or those done for commercial purposes [11]. For many others research involving human cloning and genetic engineering in general were also unacceptable [10,12,56]. Finally, some respondents opposed research done for a commercial profit [4,6,12,14]. Nevertheless, apart from these concerns, most respondents believed that the benefits outweigh the risks [22,51,54].
Point 6:
185 3.6. Preferred Type of Consent
Many types of informed consent (IC) mentioned: Blanket, broad, dynamic, tiered, general, narrow, all these IC must be explained and cited properly
Response 6: The paragraph was revised and more explanations were added.
Informed consent in the context of biobanking is a hotly discussed ethical and societal issue. Classic consent (specific or narrow, i.e. consent for one experiment with well-defined purposes, risks and benefits) is not possible due to objective reasons - biospecimens are used in many researches, by many scientists working at different places. Therefore, new models of consent are proposed, such as blanket consent (it refers to a process by which individuals donate their samples without any restrictions), broad consent (refers to a process by which individuals donate their samples for a broad range of unspecified future studies with some restrictions), dynamic consent (a digital decision-support where modern IT communication strategies are used to continuously inform and offer choices to donors to specify the types of research for which their specimens can be used or not) or tiered consent (research can be subdivided into tiers or categories and participants can specify the types of research for which their specimens will be used). Although from a biobank’s perspective broad consent is preferable [69], it is not optimal for the donors. Thus, while some studies indicate respondents’ preferences for broad consent [12,29,38,43,45], others show that, if available, other options are preferable [4]. For example, while in Europe, general consent was more accepted in Scandinavia (33%-42%) than in Southern countries: Bulgaria, Greece and Turkey (11%-16%), 67% of European population opted for narrow consent while and only 24% preferred broad consent [8]. Moreover, most Swedes rejected the possibility of conducting any type of research without explicit consent of the donors (62%), and 22.3% wanted to re-consent for every new research [18]. Also, 42% of Finns expected re-consent, if a new research differs from the original, and 29% before any new research [35]. 60% of Canadian leukemia patients preferred broad consent but 30% chose tiered consent and 10% – specific consent [42]. In another American study, 44% of respondents perceived broad consent as unacceptable and 38% as the worst option. Interestingly, for 43%, specific consent was also unacceptable and for 45% it was the worst type of consent [46]. Thus, while the donors reject both extremes, they want to preserve some control over the donated samples [14,66].
Point 7:
223 3.8. Demographic Characteristics of Potential Donors
Middle-aged, older persons young, please specify especially for „young“ there are different „age“ cathegories, in some countries it is 16, in others 18,…
Response 7: No study has measured the attitudes from respondents under 16 years of age; thus, in most studies analysed young meant under 30 years old, middle age was between 40 and 65, while older were usually defined as those aged over 65. To clarify we have added some additional data and now the paragraph states as follows:
Some studies suggest that middle-aged (usually 40-65 years old) persons and older are more favourable toward donation, trust biobanks and accept broad consent more often [11,26,29,30,43]. For example, a study by Lewis et al. showed that respondents over 55 years of age were more eager to donate [11] while in another research conducted by Goddard et al almost 70% of persons over forty were in favor of donation. In contrast, only 31% of those under that age would donate [29]. On the other hand, some studies suggest that individuals aged under fifty were more in favor of specific consent [4, 27] and that increasing age reduces the number of respondents willing to donate [7,32,38].
References:
In following references, I recommend revisions:
Point 8:
5. why „ithalics“, the same 20., 63.,
Response 8: There was no italics in the original version of the manuscript. Some error must have occurred. It has been corrected.
Point 9:
24. must be revised
Response 9: IRB Ethics and Human Research has been changed to IRB Ethics Hum. Res
Point 10: 48., 66. how many authors do you have to mention?
Response 10: As the Instructions for Authors does not contain any direct information how many authors have to be mentioned in references, we have decided to list all of them. At the same time, we are aware that some papers published in IJERPH list only 9, 10 or 11 authors with additional el al.
Of course, if we receive information how many authors should be mentioned, we will revise those two references.
Reviewer 2 Report
I have read with great interest the paper by Domaradzki and Pawlikowski which focuses on an important field such as that of citizen involvement in biobanking. The first point that clearly emerges is the lack of awareness of biobanks existence, suggesting the need of a wider public communication, considering all donor’s perceived benefits and risks of biobanking in order to build a trust towards research that represents the prerequisite for participation. The second point that transpires is the fear that personal data might be disclosed by joining to a biobank research protocol. Indeed, the matter of security is strongly felt and laws are continuously updated in order to ensure the security of personal data which are subject to continuous dependent variations by the novel knowledge acquired due to technical progress and by the nature of the data themselves. Again, it is of outmost importance to reassure the donors that their personal data can only be used to the extent that is strictly necessary for the achievement of specific objectives, which will have to be identified and disclosed to the person. On the whole, I appreciate this effort and have no concern regarding the manuscript.
Author Response
Dear Madam/Sir,
We sincerely acknowledge all your constructive and insightful suggestions for the improvement of our paper on Public Attitudes Toward Biobanking of Human Biological Material for Research Purposes.
We have applied all Your suggestions which You can find both in the above Cover Letter and in the attached file with the revised manuscript. To facilitate tracking, all changes have been marked red, while additional comments can be traced by using the "Track Changes" function in Microsoft Word.
Yours Sincerely
Jan Domaradzki